

# Insight into the structure of black coatings of ancient Egyptian mummies by advanced Electron magnetic resonance of vanadyl complexes.

Charles E. Dutoit,[1] Laurent Binet,[1] Hervé Vezin,[2] Océane Anduze,[1] Agnès Lattuati-Derieux,[3] Didier Gourier [1]

[1] Chimie-ParisTech, PSL University, CNRS, Institut de Recherche de Chimie-Paris (IRCP), F-75005 Paris, France

[3] Centre de Recherche et de Restauration des Musées de France (C2RMF), Palais du Louvre, F-75001 Paris, France

[2] Université de Lille, CNRS, UMR8516-LASIRE, F-59000 Lille, France.

*Correspondence to* : Didier Gourier (didier.gourier@chimieparistech.psl.eu)

**Abstract.** Ancient Egyptian mummies from the Late Period to the Greco-Roman period were covered by a black coating consisting in a complex and heterogeneous mixtures of conifer resins, wax, fat and oil with variable amounts of bitumen. Natural bitumen always contains traces of vanadyl porphyrin complexes that we used here as internal probes to explore the nanoscale environment of $V^{4+}$ ions in these black coatings by Electron Nuclear Double Resonance (ENDOR) and Hyperfine Sublevel CORrelation spectroscopy (HYSCORE). Four types of vanadyl porphyrin complexes were identified from the analysis of $^{14}N$ hyperfine interactions. Three types (referred to as VO-P1, VO-P2 and VO-P3) are present in natural bitumen from the Dead Sea, among which VO-P1 and VO-P2 are also present in black coatings of mummies. The absence of VO-P3 in mummies, which is replaced by another complex VO-P4, may be due to its transformation during preparation of the black matter for embalming. Analysis of $^{1}H$ hyperfine interaction shows that bitumen and other natural substances are intimately mixed in these black coatings, with aggregate sizes of bitumen increasing with the bitumen content, but not exceeding a few nanometres.

## 1 Introduction

Mummies and wooden coffins, funerary artifacts and panel paintings in ancient Egypt were often covered with organic black materials, made of a heterogeneous mixture of natural substances such as fat, oil, wax, conifer or mastic tree resin, pitch, animal glue, plant gum and bitumen in variable proportions (Maurer, et al., 2002; Buckley, et al., 2004; Clark, et al., 2016; Fulcher, et al., 2021). All these components are characterized by a variety of molecular biomarkers identified mainly by gas chromatography – mass spectrometry (GC-MS) analysis. However, the presence or not of bitumen in these black materials has been the subject of a controversy in the past due to the fact that the analytical protocols used were often not well-adapted to the detection of bitumen, so that two opposite opinions have emerged among researchers analyzing these black coatings: those who are doubtful about the presence of bitumen (Lucas and Harris, 1989; Buckley and Evershed, 2001; Davis, 2011), and





those who claimed its presence (Spielmann, 1933; Rullkötter and Nissenbaum, 1988; Connan and Dessort, 1989; Colombini, et al., 2000; Harrell and Lewan, 2002; Maurer, et al., 2002). Thanks to the identification of specific biomarkers (hopanes, steranes) and radiocarbon analyses (bitumen has lost its $^{14}C$), a consensus has emerged on the increasing presence of bitumen in embalming materials from the New Kingdom (ca 1550-1070 BC) to the Ptolemaïc/Roman period ending in the 4th century AD (Clark, et al., 2016). Despite the inestimable contribution of GC-MS for revealing the molecular composition of these black coatings, this micro-destructive technique requires preliminary steps of fractionation and separation, which exclude any direct, identification of bitumen, so that structural information on this black material cannot be obtained at the nanometer scale. This type of information requires the use of non-destructive techniques, which preserve the micro/nanostructure of the material, *i.e*. which leave the samples intact. This is the case with magnetic resonance techniques because the low frequency electromagnetic fields (radiofrequency for NMR and microwave frequency for EPR) penetrate the whole sample and deposit a negligible energy in the material compared to the other spectroscopic techniques. Multinuclear magnetic resonance ($^{1}H$- and $^{23}Na$-NMR) of mummified tissues is mainly used in the imaging mode (Mûnnemann, et al., 2007; Özen, et al., 2016) rather than the spectroscopic mode (Karlik, et al., 2007), owing to the rather low sensitivity and spectral resolution of NMR for these highly disordered solid materials. Electron paramagnetic resonance (EPR) is the electronic equivalent of NMR, and applies in the presence of unpaired electron spin density, *i.e* with electron spin $S \geq 1/2$. The spectroscopic resolution of EPR is optimal when the paramagnetic entities (transition metal ions, radicals, …) are magnetically diluted, which correspond to defects and impurities in material (Bertrand, 2020). It is well known that oil and bitumen contain organic radicals and porphyrinic complexes of vanadyl $VO^{2+}$ ions ($V^{4+}$ ion, $3d^1$ configuration). These very stable paramagnetic molecules are present mainly in oil and bitumen, and in asphaltene - the most refractory fraction of oil and bitumen – and can be considered as molecular markers of bitumen which can be detected with high sensitivity by EPR spectroscopy (Rullkötter, et al., 1985; Baker and Louda, 1986; Premovic, et al., 1998; Ben Tayeb, et al., 2015). Generally speaking, vanadyl porphyrin complexes (hereafter referred to as VO-P) are specific of oils and bitumen of marine origin (Barwise, 1990; Breit and Wanty, 1991; Lopez and Lo Monaco, 2017), while carbonaceous radicals (hereafter referred to as $C^0$) are present in all fossilized organic matters, whether of marine or terrestrial origin (Uebersfeld, et al., 1954; Skzypczak-Bonduelle, et al., 2008; Bourbin, et al., 2013), and even in the extraterrestrial carbonaceous matter (kerogen) of carbonaceous meteorites, the most primitive objects of the solar system (Binet, et al., 2002). Recently, we showed that EPR analysis of VO-P complexes and $C^0$ radicals is a simple and nondestructive way (no sample preparation) to reveal the presence of bitumen in black coatings of Egyptian mummies, even in very small amount (Dutoit, et al., 2020). In addition to VO-P, these black coatings contain also non-porphyrinic $VO^{2+}$ complexes (hereafter referred to as VO-nP), with four oxygen ligands in nearly square planar configuration (Dutoit, et al., 2020). These VO-nP complexes are absent in the Dead Sea bitumen used by Egyptians, and are found in black coatings containing natural substances in addition to bitumen. We hypothesized that these VO-nP could be localized at the interface between bitumen and other natural substances, and result from the de-metalation of VO-P of the former followed by the complexation of $VO^{2+}$ by oxygenated functions of the latter (Dutoit, et al., 2020).



However the resolution of EPR spectra of VO-P in such highly disordered materials is limited by the fact that the weak hyperfine (hf) interactions with other nuclei, namely $^1H$ (I=1/2, 100% abundance), $^{14}N$ (I = 1, 99.6 % abundance) and $^{13}C$ (I = 1/2, 1.1 % natural abundance) are unresolved. These hindered hf interactions contain precious information about the structure

of VO-P complexes, their environment and possibly on the degree of alteration of the black matters. This information can be recovered by the indirect detection of NMR transitions of magnetic nuclei in the environment of the unpaired electron spin. By this means, $VO^{2+}$ of bitumen can be considered as internal probes which "see" their nuclear spin environment in a non-destructive manner. Here we used Electron Nuclear Double Resonance spectroscopy in continuous-wave mode (cw-ENDOR) for $^1H$ nuclei, and HYperfine Sublevel CORrelation (HYSCORE) spectroscopy for $^{14}N$ nuclei to study the same corpus of

Egyptian black coatings than in the preliminary cw-EPR study (Dutoit, et al., 2020). We found that proton ENDOR is sensitive to the amount of bitumen in the black matters and to the size of bitumen aggregates, while HYSCORE of nitrogen reveals the presence of different types of VO-P complexes in the material. Special attention was given to the black coating of a human mummy of unknown origin (**Hum 3**), but whose EPR characteristics differ clearly from coatings of the other studied mummies (Dutoit, et al., 2020).


## 2 Experimental procedures

### 2.1 Samples.

The artifacts and mummies from which the samples of black coatings were collected, as well as their origin are described in Fig. 1 and Table 1, and in more details in Table S1. The collected samples are shown in Fig. S1 in Supplementary information

(SI). Three fragments were taken from the coating of an anthropomorphic coffin (labelled **Hum 1**) dated from the Ptolemaïc period (332 BC-30 BC), and two human mummies: **Hum 2** (end of the IV$^{th}$ century BC) and **Hum 3** (presumably XXV$^{th}$ dynasty, 744 BC-656 BC). Four fragments were taken from animal mummies dated from the same periods as human mummies, among which three rams **An 1**, **An 2**, **An 3** dating from the Late period (664 BC – 322 BC) and one crocodile **An 4** (Ptolemaïc period 332 BC- 30 BC). Their EPR spectra were compared with those of two pure bitumen samples: a fragment of natural

asphalt from the Dead Sea (**Ref 1**), and a commercial powder of bitumen of Judea (**Ref 2**). Dead Sea asphalt was by far the most important source of bitumen supply in Egypt for the period corresponding to the samples studied in this work (Fulcher, et al. 2021). The commercial Judean bitumen (**Ref 2**) having undergone preparatory treatments (undocumented), it allows to test the stability of paramagnetic species, and thus their relevance as a proxy for the study of black coatings. All samples (10-20 mg, Fig.S1) were inserted into quartz Suprasil EPR tubes.

### 2.2 EPR, ENDOR and HYSCORE experiments.

Continuous wave Electron Paramagnetic Resonance (cw EPR) measurements were performed at room temperature and at 100 K with a Bruker Elexsys E500 EPR/ENDOR spectrometer operating at about 9.6 GHz (X band) and 34 GHz (Q band), equipped with a high sensitivity X band 4122SHQE/0111 EPR cavity and a Q band ER5106QTE resonator for both EPR and ENDOR. Cw ENDOR at Q-band was used to measure the hyperfine interaction with $^1H$ nuclei of porphyrin ligands and their



molecular environment (see Fig. 3a). The ENDOR spectra were recorded at 100K by using a CF935 helium flow cryostat from Oxford Instruments. The radio-waves were amplified by an ENI3100L amplifier, and the ENDOR signals were detected by a 25 kHz frequency modulation of the sweeping rf field, with a modulation depth of 100 kHz. The rf was swept in the range 45-60 MHz, centered at the proton Larmor frequency.

Pulse EPR experiments at X-band were carried out with a Bruker ELEXSYS E500 spectrometer equipped with a Bruker
cryostat "cryofree" system. Two-pulse echo field sweep EPR spectra were recorded with the standard Hahn echo sequence $\pi/2$-$\tau$-$\pi$-$\tau$-echo. The resulting echo-detected absorption EPR spectrum (ED EPR) was pseudo-modulated to give a first derivative ED EPR spectrum similar to the cw EPR spectrum.

HYSCORE experiments were performed at 6K with the pulse sequence $\pi/2$-$\tau$-$\pi/2$-t1-$\pi$-t2-$\pi/2$-$\tau$-echo, with pulse lengths of 22 ns and 44 ns for $\pi/2$ and $\pi$ pulses, respectively, and the delay $\tau = 200$ ns was chosen as an optimum to prevent blind spot effects
(Ben Tayeb, et al., 2015). The spectra were recorded with 256×256 data points for t1 and t2 time domains. The unmodulated part of the echo was removed by second-order polynomial subtraction. Final HYSCORE spectra were obtained by 2D-Fourier transformation of the data set, using a Hamming apodization window function. EPR and HYSCORE spectra were simulated with the EasySpin toolbox for Matlab (version 5.2.28) (Stoll and Schweiger, 2006).

**Table 1.** Origin of the samples of black matters

| Label | Origin |
|---|---|
| *Hum 1* | Anthropomorphic coffin, upper Egypt, Ptolemaïc period (332 BC-30 BC). Black coating in the bottom of the coffin |
| *Hum 2* | Human mummy, Egypt, late Period (end of the IV[th] century BC); black coating covering the mummy |
| *Hum 3* | Human mummy, Egypt, Late Period (XXI[th] -XXV[th] dynasty). Black matter taken from the neck of the mummy. |
| *An 1* | Ram mummy, Upper Egypt, Late Period; 672 BC–322 BC; black coating covering the mummy. |
| *An 2* | Ram mummy, Upper Egypt, Late Period (664 BC–332 BC); black coating covering the mummy |
| *An 3* | Ram mummy (the same as **An 2**); tissue with brown matter covering the mummy. |
| *An 4* | Crocodile mummy, Upper Egypt, Ptolemaïc period; black matter covering the skull. |
| *Ref 1* | Fragments of natural asphalt from the Dead Sea |
| *Ref 2* | Commercial powder of bitumen of Judea |





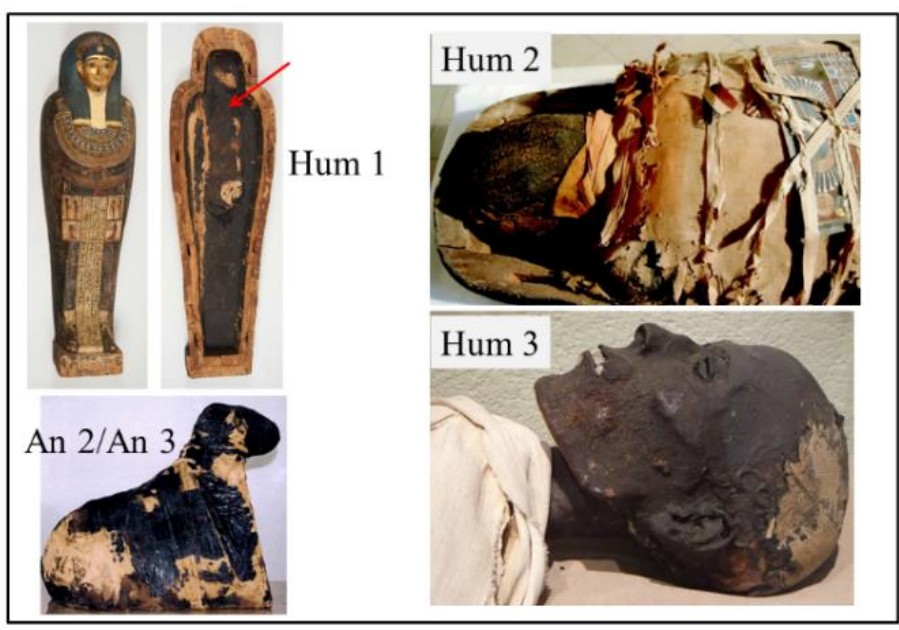

**Figure 1.** Origin of some samples of black matters studied by EPR; ***Hum 1***) Coffin of *Irethorerou* (Ptolemaïc dynasty), *The Art and History Museum, Narbonne, France* (C2RMF76267, images cha17156 and cha17158); the black matter was sampled at the bottom of the coffin (arrow); © C2RMF/Anne Chauvet; ***Hum 2***) Human mummy of the Late Period, *The Hieron museum, Paray-le-Monial, France*, the sample is a fragment of black matter covering the mummy, © Hélène Guichard; ***Hum 3***) Head of the mummy from the Late period, *Chateau-musée, Boulogne, France,* the sample was taken from the mummy's neck, reproduced with permission, © Frédérique Vincent; ***An 2/An 3***) Ram mummy of the Late Period, *The Thomas Dobrée museum, Nantes, France* (Inv. D 961.2.140); ***An2*** and ***An3*** samples are a fragment of black matter and a fragment of tissue with brown matter, respectively, © Dépôt du musée du Louvre / Musée Dobrée-Grand Patrimoine de Loire-Atlantique.

## 3 Results and discussion

### 3.1 EPR spectra.

Examples of cw EPR spectra at X band and Q band are shown in Fig.2, for the Dead Sea asphalt (***Ref 1***) and the black coatings of mummies ***An 2*** and ***Hum 3***. The other spectra are given in Supplementary Information Figs. S2-S4. All spectra show the two paramagnetic species classically present in bitumen and oil (Saraceno, et al., 1961; Aizenshtat, et al., 1989; Premovic, et al., 1998; Ben Tayeb, et al., 2015), and recently identified by X band EPR in black coatings of mummies (Dutoit, et al., 2020): (i) organic radicals $C^0$ of asphaltene, represented by a single and intense line in the central part of the spectrum, with $g = 1.9994$, and (ii) the spectrum of vanadyl porphyrin (VO-P) complexes embedded in the asphaltene, resulting from the hyperfine (hf) interaction of the $S = 1/2$ electron spin of $V^{4+}$ (configuration $3d^1$) with the $I = 7/2$ nuclear spin of the 100% abundant $^{51}V$





isotope. This hf interaction gives two sets of $2I+1 = 8$ lines characterized by $g$ and hf parameters $A$ given in Table 2. Parameters $g_{//}$ and $A_{//}$ correspond to VO-P complexes with the V-O bond oriented along the external field $\mathbf{B}_0$. The most intense central set of 8 lines, with parameters $g_\perp$ and $A_\perp$ corresponds to VO-P complexes oriented with $\mathbf{B}_0$ lying in the porphyrin plane (and thus

perpendicular to the VO bond). The $^{51}$V hf lines probed in HYSCORE (at X band) and ENDOR (at Q band) experiments are represented by arrows in Fig.2a and 2b, respectively.

In addition, the black coatings of **An 2** (Fig.2), **Hum 1**, **2** and **An 1, 3, 4** samples exhibit additional lines (some of them are marked by green circles in Fig.2 and Figs. S2 and S3) characterized by $g_{//} = 1.925\pm0.003$, $g_\perp = 1.978\pm0.003$, $A_{//} = 176(\pm3)\times10^{-4}$ cm$^{-1}$ and $A_\perp = 70\pm(3)\times10^{-4}$ cm$^{-1}$ and attributed to non-porphyrinic vanadyl complexes with oxygenated ligands in nearly

square planar configuration (referred to as VO-nP complexes) (Dutoit, et al., 2020). These complexes are thought to result from the interaction of bitumen with other bioorganic substances (resins, wax, fat …) of the black matters. The only exception is **Hum 3** sample (the human mummy from the Boulogne museum, Fig.1), which exhibits only the VO-P spectrum, like pure bitumen **Ref 1** (Fig.2) and **Ref 2** (Fig.S2a). We concluded that mummy **Hum 3** was covered with pure bitumen, which was confirmed by GC-MS analysis (Dutoit, et al., 2020). In addition, the spectrum of **Hum 3**, **An 1** (Fig. S2b,c) and, to a lesser

extent **An 2**, show a broad baseline distortion due to a ferromagnetic resonance (FMR) signal of iron oxide microparticles. This FMR signal, which does not give electron spin echo, can be eliminated by recording the echo-detected EPR spectrum (ED EPR), as clearly shown for **An 2** and **Hum 3** at X band (Fig. 2a).

The EPR parameters of VO-P complexes were deduced from the fitting of spectra at both X and Q bands (Table 2). The slight differences of EPR parameters between the three samples fall within error bars of the simulations, except for parameter $A_\perp$, for

which the differences are clearly visible on the spectra at Q band (Fig. 2b).

The shape of EPR spectra of VO-P complexes (Fig.2) is entirely controlled by the anisotropies of the $g$-factor and of the strong hf interaction with the central $^{51}$V nucleus, which reflect the electronic structure and the geometry of the complex. For this reason, the weak unresolved hf interactions with $^{1}$H, $^{14}$N and $^{13}$C nuclei of the porphyrin ligand can only be revealed by hyperfine spectroscopy.

**Table 2.** EPR parameters of VO-P complexes in **Ref 1**, **An 2** and **Hum 3** samples

| Sample | $g$-factors ($\pm 0.002$) | $A$ (in $10^{-4}$ cm$^{-1}$) |
|---|---|---|
| **Ref 1** | $g_{//} = 1.959$ | $A_{//} = 157 \pm 3$ |
|  | $g_\perp = 1.980$ | $A_\perp = 54 \pm 2$ |
| **An 2** | $g_{//} = 1.956$ | $A_{//} = 158 \pm 3$ |
|  | $g_\perp = 1.977$ | $A_\perp = 55 \pm 2$ |
| **Hum 3** | $g_{//} = 1.957$ | $A_{//} = 155 \pm 3$ |
|  | $g_\perp = 1.978$ | $A_\perp = 52 \pm 2$ |



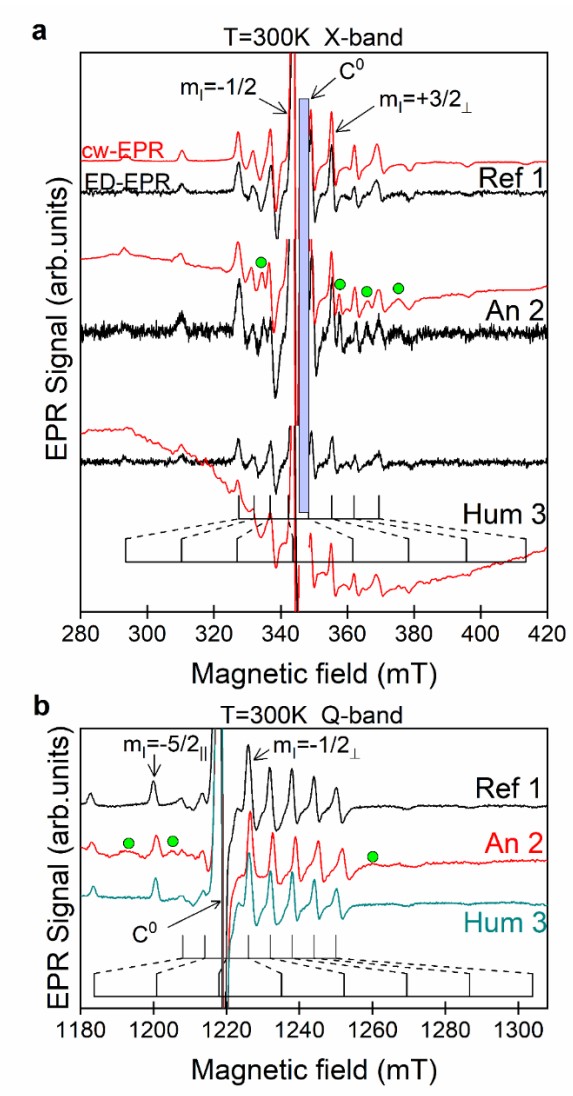

**Figure 2**. EPR spectra at room temperature of the black matters from samples *Ref 1*, *An 2* and *Hum 3*; a) cw EPR spectra (in red) and pseudo-modulated ED EPR spectra (in black) at X band; the vertical blue area represents the position of the sharp $C^0$ signal, that has been suppressed for the sake of clarity; b) cw EPR spectra at Q band; Some EPR lines of VO-nP complexes in *An 2* are represented by green circles. The magnetic field settings for ENDOR (at Q band) and HYSCORE (at X band) experiments correspond to [51]V hyperfine lines marked by arrows.



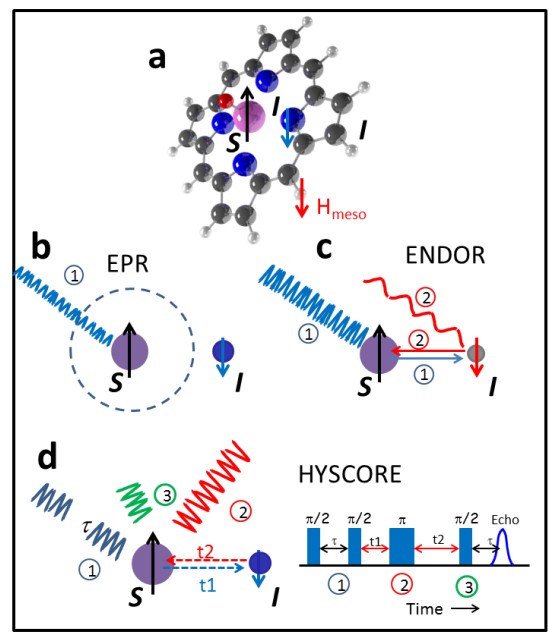

**Figure 3.** Principle of cw ENDOR and HYSCORE spectroscopy; a) Schematic structure of a porphyrin vanadyl complex, showing the electron spin $S$ on vanadium and two nuclear spins $I$ on nitrogen and on the bridging hydrogen $H_{meso}$; b) EPR with a non-saturating microwave field, the dashed circle representing the limited resolution of EPR, which does not reach neighboring ligand nuclei; c) cw ENDOR: a saturating microwave field modifies the populations of the nuclear spin state (step 1), while a strong rf field at nuclear frequency restores the nuclear populations, which in turn desaturates the EPR transition (step 2); d) HYSCORE spectroscopy: a sequence of two $\pi/2$ microwave pulses separated by time $\tau$ induces a nuclear coherence in each electron spin state $m_s$ (step 1), a $\pi$ pulse after time t1 produces a transfer of nuclear coherence between the two electron spin states $m_s$ (step 2); after an evolution time t2, the nuclear coherences are transferred back to the electron coherence by a $\pi/2$ pulse, giving an electron spin echo at time $\tau$ (step 3).

## 3.2 $^1$H ENDOR analysis.

EPR spectra are recorded with weak, non-saturating microwave radiation (Fig.3a,b). In a cw ENDOR experiment, an EPR transition is partially saturated at high microwave power and at fixed magnetic field, which modify the population of the nuclear spin states. A saturating radiofrequency (rf) field of frequency $\nu$ is then swept through the NMR frequencies of $^1$H nuclei (Fig.3c). The populations of nuclear spin states are modified at each nuclear resonance frequency, which are detected by a small increase in the EPR intensity (ENDOR enhancement). A typical $^1$H ENDOR spectrum at Q band of VO-P complex in pure bitumen ***Ref 2*** is shown in Fig.4a, recorded at observing fields corresponding to the $m_I = -1/2_\perp$ and $-5/2_{//}$ $^{51}$V-hf lines (arrows in Fig.3b). The $^1$H signal is centered on the Larmor frequency $\nu_H$ of hydrogen (typically $\nu_H = 51.9$ MHz for a magnetic field 1226 mT). As ENDOR spectra are recorded as first derivative of the ENDOR enhancement with respect to the rf



frequency, the spectral features occur at angular turning points characterized by $\nu - \nu_H = \pm A_{//}/2$ and $\nu - \nu_H = \pm A_\perp/2$, where $A_{//}$ ($A_\perp$) correspond to the hf parameters of hydrogen atoms in VO-P complexes oriented such that the external field $\mathbf{B}_0$ is parallel (perpendicular) to the V…H directions. The shape and parameters of the ENDOR spectrum depend on the nature of the C-H bond, on the V…H distance, and on the set of molecular orientations selected by the magnetic field settings. For an observing

field set at the $-5/2_{//}$ EPR transition of VO-P, this corresponds to the selection of vanadyl complexes oriented with the V-O bond nearly parallel to the field vector $\mathbf{B}_0$, which is thus perpendicular to all the V…$H_{meso}$ directions of the porphyrin plane (see Fig.2a) so that only two ENDOR lines corresponding to $A_\perp$ are observed for these hydrogens (Fig.4a, bottom). Alternatively, for a field setting at the $-1/2_\perp$ EPR transition of VO-P, the selected vanadyl complexes correspond to $\mathbf{B}_0$ lying in the porphyrin plane and thus covering all possible angles with V…$H_{meso}$ directions, so that both $A_{//}$ and $A_\perp$ ENDOR lines are

observed in this case (Fig.4a, top). The values $A_{//} = 2.6$ MHz and $A_\perp = 0.3$ MHz measured for VO-P correspond to the values known for the bridging C-$H_{meso}$ bond of porphyrin in VO-P complexes (Biktagirov, et al., 2017; Manniko and Stoll, 2019).

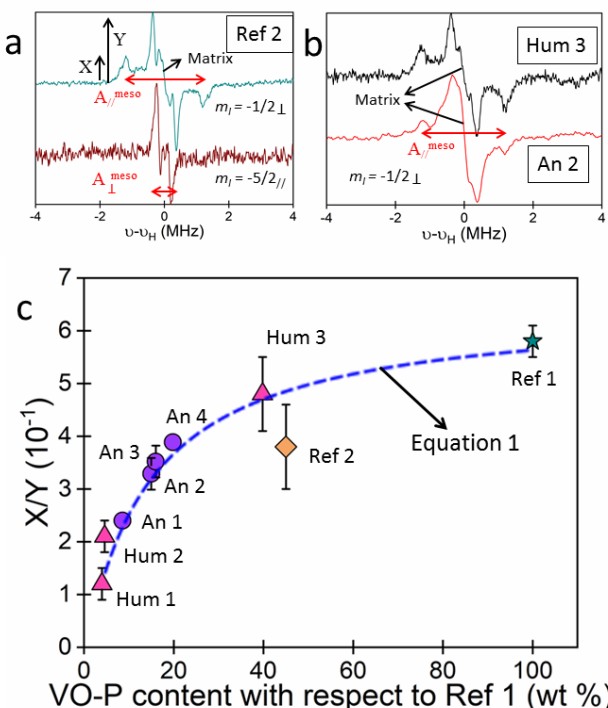

**Figure 4**. $^1$H ENDOR at Q band and at 100K of VO-P complexes. a) bitumen of Judea (**Ref 2**), recorded by observing the $m_I = -1/2_\perp$ and $-5/2_{//}$ hf lines of $^{51}$V (shown in Fig.2b); b) **Hum 3** and **An 2** samples recorded at the $m_I = -1/2_\perp$ hf position; c)

Variation of the ratio X/Y of ENDOR amplitudes versus the weight-normalized VO-P content of samples; The interrupted line was calculated from Eq.1 with $b = 15$.





It must be noticed that additional ENDOR features that would be expected at $A_{//} = 1.3$ MHz and $A_{\perp} = 0.4$ MHz for hydrogen of pyrrole groups (Gourier, et al., 2010; Manniko and Stoll, 2019), are not observed in our ENDOR spectra. This indicates that

all pyrrolic hydrogen atoms are substituted with alkyl groups, as known in the case of vanadyl geoporphyrins in oil and bitumen, such as vanadyl etioporphyrin (VO-EP) and vanadyl deoxophylloerythroetioporphyrin (VO-DPEP) for example (Fig.S6) (Dechaine and Gray, 2010). The narrow signal centered at $\nu_H$, referred to as the "matrix" line (Kevan and Kispert, 1976), corresponds to more distant hydrogen atoms, which are weakly coupled to the electron spin by dipolar interactions. These include hydrogens of the alkyl side groups linked to pyrroles (mainly methyl and ethyl groups, see Fig. S6) and to

hydrogens of the polyaromatic molecules of the surrounding asphaltene molecules, but also to hydrogens of the bioorganic molecules (resins, waxes and fats) of the black matters. In the case of pure bitumen (***Ref 1*** and ***Ref 2***, see Fig.4a and Fig. S5), only the alkyl side groups of porphyrin ligands and the asphaltene hydrogens contribute to the weak matrix ENDOR line.

Although EPR spectra of VO-P complexes are very similar in bitumen and all black coating samples (Figs. 2b and S4), their ENDOR spectra are somewhat different. The $A_{//}$ lines of $H_{meso}$ atoms are well resolved in pure bitumen ***Ref 1***, ***Ref 2*** and in the

human mummy ***Hum 3*** (Fig.4b) while they broaden and decrease in intensity in black coatings of other samples (Figs.4b and S5). Moreover, a pronounced matrix ENDOR signal centered at $\nu_H$ appears in all samples except pure bitumen (***Ref 1*** and ***2***) and ***Hum 3*** (Figs.4b and S5).

This variation of the $^1$H ENDOR spectrum can be quantified by the ratio X/Y, where X is the amplitude of the spectrum at the position of the $A_{//}$ line, and Y the amplitude at the position of the $A_{\perp}$ line, including the matrix line (see Fig. 4a). The variation

of X/Y with VO-P content of the black coatings, normalized to the same mass of asphalt from the Dead Sea (***Ref 1***) is shown in Fig. 4c. We observed a regular decrease of X/Y with decreasing content of VO-P per unit mass of black matter, the VO-P content being related to the bitumen content. A value X/Y ≈ 5.6 is found for the pure natural bitumen (***Ref 1***), where the environment of VO-Ps consists mostly in asphaltene molecules. Mixing bitumen with increasing amounts of natural substances has three effects: (i) the decrease of VO-P content per unit mass of black matter (*i.e* the decrease of EPR intensity of VO-P);

(ii) an increase of the matrix ENDOR line (*i.e* an increase of Y) due to increasing amounts of hydrogen atoms of bioorganic molecules in the vicinity of VO-P, and; (iii) a broadening and weakening of the $A_{//}$ ENDOR lines of hydrogen $H_{meso}$ (*i.e.* a decrease of their amplitude X) resulting from a disorder (distribution of hf interaction values $A_{//}$) in the structure and environment of VO-P complexes. However, this disorder effect is not sufficient to modify the global square planar geometry of VO-P, and has no visible effect on the shape and parameters of EPR spectra. The reference bitumen ***Ref 2*** deviates from the

general trend in Fig.4c. However, since this commercial bitumen has been processed, it may have undergone some undocumented chemical/physical treatment that could have modified its micro/nano structure. For this reason, the data for ***Ref 2*** has been plotted in Fig.4c only for the sake of comparison with historical and geological samples.

It is important to note that a matrix ENDOR line can occur only if the vanadium-hydrogen distances $R$ are sufficiently small to give non-zero electron-proton dipolar interactions (*i.e* $R < 5$-6 nm) (Kevan and Kispert, 1976; Stoll, et al., 2005). We may

conclude that only protons at distance $R > 0.6$-0.7 nm from the vanadium atom (~ half the size of the porphyrin ligand) and $R$



< 5-6 nm (limit for non-zero dipolar interaction) can contribute to the $^1$H matrix line. This limited distance range for matrix protons has two consequences: (i) for a given proportion of bitumen and natural bioorganic substances, the amplitude $Y$ of the matrix line should increase ($X/Y$ decrease) upon decreasing the radius $R_A$ of bitumen aggregates in the black matters (see Fig.S7 and section 4 of the SI for details on the model), and reach a maximum amplitude for $R_A$ < 5-6 nm (*i.e.* all VO-P complexes

"see" hydrogens of natural bioorganic substances); (ii) for bitumen aggregates with mean $R_A$ value, $X/Y$ should decrease upon increasing content of natural substances, as more and more bioorganic hydrogen atoms are present in the vicinity of VO-P complexes. According to (i), $X/Y$ should be nearly independent of VO-P content of the black matter if bitumen aggregates are large ($R_A$ in the micrometer range or larger), because in this case, only the small fraction of VO-P close to the surface of bitumen aggregates "see" the bioorganic protons. The vast majority of VO-Ps being located in the volume of the bitumen

aggregates, they should be insensitive to the presence or not of biomolecules around these aggregates because the hydrogens are at too great a distance to contribute to the matrix line. In this case the ratio $X/Y$ should be nearly independent of the VO-P content of the samples and should be close to the value 0.6 for **Ref 1**, which is not observed experimentally. According to (i) and (ii), the fact that $X/Y$ decreases regularly with decreasing VO-P content (Fig. 4c) for our body of black coatings (if we except the commercial bitumen **Ref 2**) indicates that the bitumen aggregates are very small ($R_A$ < 6-7 nm) in all cases, and that

$X/Y$ depends only on the ratio bitumen/natural substances of the black matters.

A lower limit of the size $R_A$ of bitumen aggregates can be estimated from the fact that VO-P and radicals C$^0$ are spatially connected in asphaltene, with (VO-P)-C$^0$ distances not larger than 1-3 nm (Mamin, et al., 2016). We previously showed that such spatial connection is conserved in bitumen of Egyptian black coatings (Dutoit, et al., 2020). Consequently, we may roughly estimate that the sizes of bitumen aggregates in the studied black coatings lie in the range ~1 nm < $R_A$ < ~6-7 nm.

Three scales of asphaltene aggregation were proposed in the Yen-Mullins model of asphaltene hierarchical structure (Mullins, 2010): the molecular (~1.5 nm), the nanocluster (~2.0 nm) and the Cluster (~5.0 nm) scales. It appears that the sizes of bitumen aggregates in black coatings correspond to the cluster scale of the Yen-Mullins model. Whatever the actual distribution size of asphaltene aggregates be, this ENDOR analysis shows that the bitumen and the other natural substances are intimately mixed in black coatings. For such small sizes of bitumen aggregates, where the surface/volume ratio is high, this could also

explain why a significant fraction of VO-P are transformed into oxygenated VO-nP complexes at interfaces between bitumen aggregates and natural substances (Dutoit, et al., 2020).

The regular variation of the ENDOR shape factor X/Y with VO-P content for all historical samples (Fig.4c) gives an information about the nanostructure of these black coatings. This variation can be simulated with a simple model considering that $X$ and $Y$ amplitudes are both sums of contributions from protons of VO-P complexes and of the biomolecular layer around

the bitumen aggregates, respectively, so that the ratio $X/Y$ is given by (see Fig.S7 and SI part 4 for demonstration):

$$\frac{X}{Y} = \frac{X_{VOP}}{Y_{VOP}} \times \frac{x + 0.048 \times b}{x + b} \tag{1}$$



where $x$ is the VOP content with respect to **Ref 1** (the abscissa in Fig. 4c), $X_{VOP}$ and $Y_{VOP}$ are the peak-heights of the parallel and perpendicular components of the [1]H ENDOR spectrum of an isolated VO-P complex (measured in **Ref 1** for $m_I = -1/2_\perp$ ). Parameter $b$ in Eq.1 is given by $b = aY_m/Y_{VOP}$, with $Y_m$ the peak-to-peak half-height of the matrix ENDOR line in each sample.

The ratio $Y_m/Y_{VOP}$ depends only on the ENDOR lineshapes. Parameter $a$ is a function of the bitumen content of the black matters and of the mean size of bitumen aggregates (see SI for demonstration). As $X_{VOP}/Y_{VOP} \approx 0.625$ for **Ref 1**, there is only one adjustable parameter $b$ in Eq.1. Experimental data were nicely fitted to Eq.1 with $b = 15$ (Fig.4c). This good agreement shows that parameter $b$ is a constant that is approximately independent of the samples (otherwise the experimental points in Fig.4c would be scattered). As developed in SI, this constant value of $b$ (except for the commercial bitumen **Ref 2**) means that

the larger the concentration of bitumen in the black matters, the larger the mean size of bitumen aggregates. This regular increase in size may be simply explained by the increasing coalescence of bitumen aggregates when they are less and less separated by the other bioorganic substances.

### 3.3 [14]N HYSCORE analysis.

The very small variations of EPR parameters $g$ and $A$ of VO-P from one sample to another (Table 2) may suggest that several types of slightly different VO-P complexes are present in variable proportions in the bitumen component of black coatings. We used HYSCORE spectroscopy at X band to discriminate different types of VO-P by their [14]N hf interaction, with the perspective to use in the future these metallic complexes for getting information on the geographical origin of the bitumen and on its chemical and thermal treatment during the preparation of mummies. In pulse EPR spectroscopy (Schweiger and Jeschke,

2001), a spin echo is generated by a series of $\pi/2$ and $\pi$ microwave pulses separated by controlled time delays (Fig.3d). By varying these time delays, the echo intensity is modulated at frequencies of the hf interactions. HYSCORE spectroscopy is based on the pulse sequence $\pi/2$-$\tau$-$\pi/2$-t1-$\pi$-t2-$\pi/2$-$\tau$-echo, where $\tau$ is the delay between the first and second $\pi/2$ pulses. The first $\pi/2$ pulse generates an electronic coherence (a mixing of the two $m_s = \pm 1/2$ states), and the second $\pi/2$ pulse after time $\tau$ transfers the electronic coherence to nuclear coherences (mixing of $m_I$ states). After an evolution time t1, a $\pi$ pulse transfers

the nuclear coherence from one $m_s$ state to the nuclear coherences of the other $m_s$ state, which creates correlations between nuclear transitions of these two $m_s$ states. After another evolution time t2, a third $\pi/2$ pulse transfers the nuclear coherence back to the electronic coherence for detection, which generates an electron spin echo after time $\tau$. The echo intensity is measured for the two times t1 and t2, which are varied stepwise at constant $\tau$ value. The 2D frequency plot (HYSCORE) is obtained by 2D Fourier transformation of the data set in time domain.

The correlations between nuclear transitions in the two $m_s = \pm1/2$ states appear as cross peaks in the 2D frequency plot, which are distributed in two different quadrants (+,+) and (+,-) corresponding to $\omega2 > 0$, $\omega1 > 0$ and $\omega2 > 0$, $\omega1 < 0$, respectively (Schweiger and Jeschke, 2001). For an electron spin $S$=1/2 interacting with an $I = 1/2$ nuclear spin such as [13]C and [1]H, we expect two cross peaks in each quadrant, which take the shape of ridges perpendicular to the diagonal $\omega1 = \omega2$ for anisotropic



hf interactions in disordered materials (Schweiger and Jeschke, 2001). The spectrum is more complicated for a nuclear spin $I$

= 1 such as $^{14}$N, which can give up to 18 cross peaks and ridges in each quadrant (Dikanov, et al. 1996). The situation is even more puzzling if the electron spin is coupled with several nuclear spins (which indeed is the case of VO-P) because such multi-spin systems can give additional zero- and multi-quanta coherences, as well as suppression effects (Stoll, et al., 2005). Fortunately, many of these spectral features are too weak to appear in the 2D plot, so that the HYSCORE spectra remain interpretable (Reijerse, et al., 1998; Garcia-Rubio, et al., 2003; Dikanov, et al., 2004). Representative HYSCORE spectra for

samples **Ref 1**, **An 2** and **Hum 3** in the frequency range 0 to ±12 MHz are shown in Fig.5. HYSCORE spectrum of **Ref 2** is reported in Fig. S8. We could not obtain acceptable HYSCORE spectra of **Hum 1**, **Hum 2** and **An 1** because of their low amount of VO-P. The spectra were recorded from two field settings in the EPR spectra (arrows in Fig.2a): the $m_I = +3/2_\perp$ hf line (Fig.5b) which select VO-P complexes oriented with their VO bond perpendicular to the external field **B$_0$** (i.e **B$_0$** lies in the porphyrin plane), and the strong $m_I = -1/2$ hf line (Fig.5a) whose position is almost independent on the field orientation (no

selection of molecular orientations). Since our goal was to explore the diversity of VO-P complexes, we focused the spectral analysis on the narrow and relatively intense correlation peaks highlighted by the black boxes in the (+,-) quadrant of Fig.5, which are represented in more details in Fig.6 for **Ref 1**, **Hum 3** and **An 2**, and in Fig.S8 for **Ref 2**.

The energy level diagram in Fig.6 describes the spin states and the corresponding nuclear transitions for an $S = 1/2$, $I = 1$ system. The frequencies for the single quantum ($\Delta m_I = 1$) and double quantum ($\Delta m_I = 2$) transitions, referred to as sq and dq

transitions, respectively, are given by (Reijerse, et al., 1998; Dikanov, et al., 2004):

$$
v_{1sq}^{\pm} = \frac{A}{2} \pm v_N + \frac{3Q}{2} + (2^{nd} \text{ order terms})
$$

$$
v_{2sq}^{\pm} = \frac{A}{2} \pm v_N - \frac{3Q}{2} + (2^{nd} \text{ order terms})
$$
(2)

$$
v_{dq}^{\pm} = A \pm 2v_N + \frac{A^{(2)}}{(A/2) \pm v_N}
$$
(3)

where $v_N$ is the nuclear Zeeman frequency, and $A$ and $Q$ the hf interaction and the quadrupolar interaction, respectively, of $^{14}$N nuclei for a given field orientation. The full expressions including second order terms in $v_{sq}$ and their estimation are given in

SI. The widths and positions of dq transitions (Eq.3) are determined by the weak anisotropy of the $^{14}$N-hf interaction $A$ and by the 2$^{nd}$ order correction, while the widths and positions of sq transitions (Eqs.2) are mostly controlled by the strong anisotropy of $Q$ (which is a traceless tensor) in addition to $A$ and 2$^{nd}$ order corrections. For this reason, the sharp dq-dq correlation peaks, which depend only on $A$ anisotropy to first order, have sufficiently high resolution and sensitivity to be used for the identification of various VO-P complexes. The quadrupolar interaction $Q$ for each type of VO-P can only be determined by

using sq frequencies (Eqs.2).



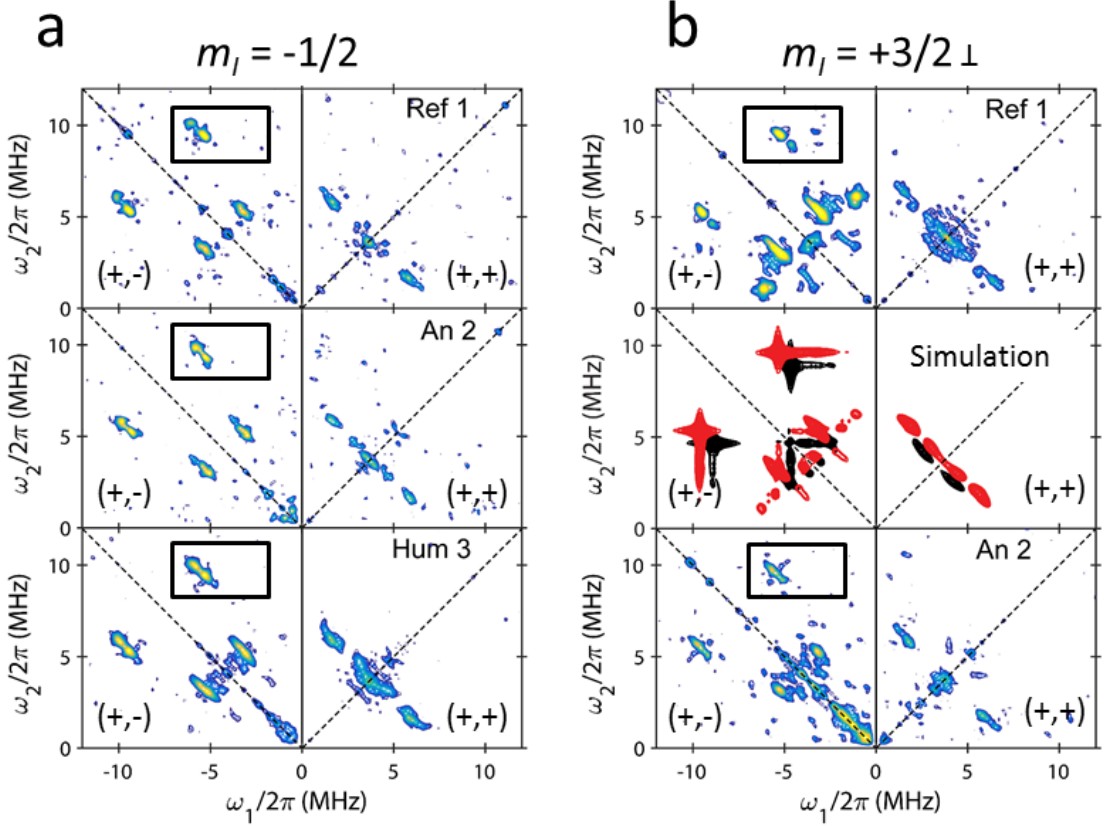

**Figure 5.** HYSCORE spectra at X band and at 6K of VO-P in Dead Sea asphalt **Ref 1** and in black coating of **An 2** and **Hum 3** mummies, recorded by observing a) the $m_I$ = -1/2 and b) the $m_I$ = +3/2⊥ hf lines marked by arrow in Fig.2a. The correlation peaks described in more details in Fig. 6 are delimited in rectangular boxes. Simulated spectra of VO-P1 (in red) and VO-P2 (in black) in the middle of panel b.

Let us take the example of the HYSCORE of Dead Sea asphalt **Ref 1**, observed at the $m_I$ = +3/2⊥ field position, and shown in the top right of Fig.6. Two distinct dq-dq peaks are clearly observed, representing two different types of VO-P complexes, hereafter referred to as VO-P1 and VO-P2. The frequency coordinates of these dq-dq peak are [-5.3; +9.5] MHz for VO-P1 and [-4.6; +8.9] MHz for VO-P2, with an error bar of about ± 0.1 MHz. The hf interaction A is directly obtained from Eq.3:

$$A = \frac{2\nu_N \left( \nu_{dq}^+ + \nu_{dq}^- \right)}{8\nu_N - \left( \nu_{dq}^+ - \nu_{dq}^- \right)} \qquad (4)$$

where the 2nd order term of Eq.3 is naturally eliminated. With $\nu_N$ = 1.1 MHz at 355.7 mT, we obtained A = - 7.28 MHz for VO-P1 and A = - 6.6 MHz for VO-P2. The hf coupling was chosen negative on the basis of DFT calculation on VO-P complex in crude oil (Gracheva, et al., 2016).





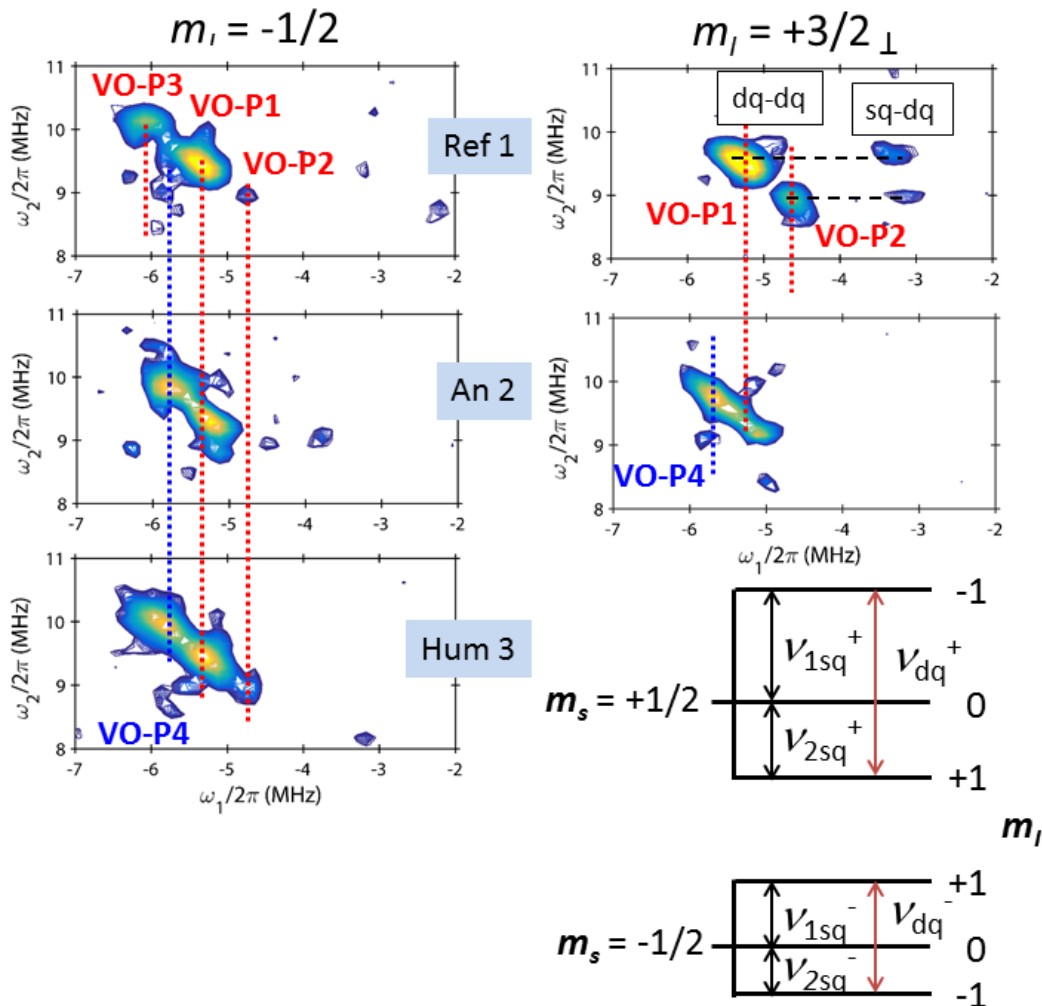

**Figure 6.** Portions of the HYSCORE spectra of **_Ref 1_**, **_An 2_** and **_Hum 3_** corresponding to dq-dq and dq-sq correlation peaks, where dq and sq represent double quantum and single quantum transitions, respectively. The diagram represents the spin states and the different nuclear transitions for a $S = 1/2$, $I = 1$ spin system.

The vanadyl content of **_Ref 1_** is sufficiently high to reveal two weaker sq-dq correlation peaks at [-3.3; +9.5] MHz for VO-P1

and [-3.1; +8.9] MHz for VO-P2, at the $m_I = +3/2_\perp$ field setting (Fig.6). As a sq-dq peak correlates a dq transition $\nu_{dq}$ of one

$m_s$ state with a sq transition $\nu_{sq}$ of the other $m_s$ state, all sq and dq frequencies of the spin diagram in Fig.6 can be simply

deduced by considering Eqs.2, which give $\nu_{1sq}^+ - \nu_{1sq}^- \approx \nu_{2sq}^+ - \nu_{2sq}^- \approx 2\nu_N$ to first order and $\nu_{1sq}^\pm + \nu_{2sq}^\pm = \nu_{dq}^\pm$. The

results for VO-P1 and VO-P2 are shown in Fig.S9. Neglecting again 2$^{nd}$ order terms, the quadrupolar parameter $Q$ can be

estimated from Eqs.2 by: $Q \approx \left(\nu_{1sq}^- - \nu_{2sq}^-\right)/3 \approx \left(\nu_{1sq}^+ - \nu_{2sq}^+\right)/3$. The values for VO-P1 and VO-P2 are $|Q| \approx 0.47 \pm 0.03$

MHz and $|Q| \approx 0.55 \pm 0.02$ MHz, respectively (see SI for the estimation of the second order terms).



The $^{14}$N hf interaction is anisotropic, with two components $A_{//}$ and $A_{\perp}$, corresponding to $\mathbf{B_0}$ parallel and perpendicular to the V-N bond. For HYSCORE spectra recorded from the $m_I = +3/2_{\perp}$ field setting, which span all orientations of $\mathbf{B_0}$ in the porphyrin plane, the measured value of $A$ is the average $\langle A \rangle_{\perp} \approx (A_{//} + A_{\perp})/2$, with $A_{//} = a_{iso} + 2T$ and $A_{\perp} = a_{iso} - T$. Parameters $a_{iso}$ and $T$ are the isotropic and dipolar hf interactions, respectively. For the corresponding HYSCORE spectrum of **Ref 1** recorded with the $m_I = -1/2$ field setting (top left spectrum of Fig.6), where almost all molecular orientations are probed, the hf interaction measured from dq-dq peaks (Eq.4) is approximated as the average value over all the possible field orientations, given by $\langle A \rangle \approx (A_{//} + 2A_{\perp})/3$. The dq-dq peak of VO-P1 at [-5.3; +9.5] MHz gives $\langle A \rangle = -7.28$ MHz for this field setting. Combining the hf values $\langle A \rangle_{\perp}$ and $\langle A \rangle$ measured for the two observing fields shows that the hf interaction is mostly isotropic, with $a_{iso} = -7.3$ MHz and $T < 0.1$ MHz for VO-P1. By the same procedure, the $^{14}$N parameters of VO-P2 are $a_{iso} = -6.5$ MHz and $T = 0.2$ MHz. It is interesting to note that the value $a_{iso} = -7.3$ MHz measured for VO-P1 is close to the value $a_{iso} = -7.23$ MHz measured by pulse ENDOR for VO-P complexes in heavy crude oil from the Republic of Tatarstan (Russia), and the value $a_{iso} = -7.2$ MHz measured by pulse EPR for vanadyl octaethylporphyrin (Fukui, et al., 1993; Gracheva, et al., 2016). This good concordance between measurements of different nature confirms that a simple measurement of the sharp dq-dq peaks recorded by observing the intense and nearly isotropic $m_I = -1/2$ EPR line, gives a good estimate of the isotropic hf coupling $\langle A \rangle \approx (A_{//} + 2A_{\perp})/3 = a_{iso}$ in VO-P complexes.

**Table 3.** Experimental (exp) and simulated (sim) hyperfine and quadrupolar parameters for $^{14}$N nuclei (*nm* = not measured) in VO-P complexes

| Complex | | $a_{iso}$ (MHz) | $T$ (MHz) | $Q_{zz}$ (MHz) | Samples |
|---|---|---|---|---|---|
| VO-P1 | exp | -7.3 | < 0.1 | 0.94 | **Ref 1**; **Ref 2**; **Hum 3**; **An 2** |
| | sim | -7.3 | 0.1 | 1.0 | |
| VO-P2 | exp | -6.5 | 0.2 | 1.1 | **Ref 1**; **Ref 2**; **Hum 3** |
| | sim | -6.6 | 0.2 | 1.0 | |
| VO-P3 | exp | (≈ -7.3) | nm | nm | **Ref 1**; **Ref 2** |
| VO-P4 | exp | -6.8 | 0.6 | nm | **Hum 3**; **An 2** |

Vanadyl complex VO-P1 is present in the four studied samples, while VO-P2 complex is detectable only in **Ref 1**, **Hum 3** and **Ref 2** (Figs. 6 and S8). Another complex, referred to as VO-P4 ($a_{iso} = -6.8$ MHz and $T = 0.6$ MHz) is also present in **An 2** and **Hum 3**, but is clearly absent in **Ref 1** and **Ref 2**. However, an additional dq-dq peak attributed to a complex VO-P3 ($a_{iso} \approx -7.3$ MHz), was detected only in pure bitumen samples **Ref 1** and **Ref 2** by observing the sharp $m_I = -1/2$ EPR transition. As sq-dq



transitions could not be detected for VO-P3 and VO-P4, it was not possible to obtain an estimation of $|Q|$ in these cases. The $^{14}$N parameters of the four VO-P complexes are reported in Table 3.

All this interpretation was based on the analysis of only a small portion of each HYSCORE spectrum (rectangular boxes in Fig.5). This procedure raises the question of the origin of all other correlation peaks and ridges present in the HYSCORE spectra (Fig. 5). To test the validity of the proposed analysis, the whole $^{14}$N HYSCORE spectra of VO-P1 and VO-P2 complexes recorded at the $m_I = +3/2\perp$ field setting were simulated with *Easyspin* software (Stoll and Schweiger, 2006), by adjusting the values of $a_{iso}$, $T$ and $Q_{zz}$ (the $z$-component of the quadrupolar interaction). The result is shown in the middle of

Fig.5b. Except for peak intensities, which are not correctly accounted for in the simulations, the main features of experimental HYSCORE spectra are well reproduced, taking into account the fact that the shape of the simulated spectra is very sensitive to the values of $a_{iso}$, $T$ and $Q_{zz}$ parameters. This validation of nitrogen parameters reported in Table 3 calls for several comments:

(i)  The simulated values $Q_{zz} = +0.9$ MHz and $+1.0$ MHz for VO-P1 and VO-P2 complexes are about twice the experimental values $|Q| = 0.47$ MHz and $0.55$ MHz measured from sq-dq correlation peaks recorded at the $m_I = +3/2\perp$ field setting. As

the quadrupolar interaction is a traceless tensor (*i.e.* $Q_{zz} + Q_{xx} + Q_{yy} = 0$), this apparent discrepancy could mean that the $z$-axis of the $^{14}$N quadrupolar interaction is nearly perpendicular to the porphyrin plane, so that the experimental values measured here by setting the magnetic field in the porphyrin plane ($m_I = +3/2\perp$ EPR transition) correspond to $|Q| \approx Q_{xx} \approx Q_{yy} \approx Q_{zz}/2$.

(ii)  All the other correlation peaks and ridges visible in the (+,-) quadrant are clearly due to the same porphyrinic nitrogen

atoms that give rise to the observed dq-dq peaks. Thus it is not necessary to invoke hf interactions with other nuclei ($^{14}$N, $^{13}$C or other).

Unfortunately the (+,+) quadrant does not give any information on the $^{13}$C hf interaction because the corresponding peaks are hindered under $^{14}$N correlations which come out in the same frequency range as $^{13}$C ($\nu_C = 3.7$ MHz), as shown by the simulation in Fig. 5. Also, in a multi-spin system such as VO-P complexes, nuclei with weak modulations (such as $^{13}$C and $^1$H) can be

partially or totally suppressed by nuclei with deep modulations, which is the case with $^{14}$N (Stoll, et al., 2005), explaining why we cannot measure $^{13}$C hf interactions in VO-P complexes.

### 3.4 Focus on the human mummy *Hum 3*.

This $^1$H and $^{14}$N hf analysis of vanadyl probes in black coatings confirms the peculiarity of **Hum 3** compared to other samples of black matter, as previously evidenced by cw EPR (Dutoit, et al., 2020). **Hum 3** was taken from the neck of the mummy

found in Nehemsimontou's coffin (XXV$^{th}$ dynasty, 744 BC to 656 BC), purchased in 1837 by the museum of Boulogne (France) from a private collector. It later turned out that the mummy and the coffin had been assembled for the purpose of a better sale, a common practice in the 19$^{th}$ century. This beautiful mummy is covered with a solid, black and shiny substance, and has therefore an unknown origin (Fig.1). Cw EPR and GC-MS analysis showed that this black coating is made of pure



bitumen (Dutoit, et al., 2020). Contrary to other black coatings studied in this work (animal and human mummies, coffin),

*Hum 3* contains no VO-nP (non-porphyrinic vanadyl complexes) (VO-nP), and its EPR spectrum is very similar to that of pure bitumen (*Ref 1* and *Ref 2*) (Dutoit, et al., 2020). This similarity with native bitumen is confirmed by the almost identical ENDOR spectra of *Hum 3*, *Ref 1* and *Ref 2* (Figs.4 and S5). These samples do not exhibit the intense $^1$H matrix line present when natural substances are mixed with bitumen, which is consistent with the fact that *Hum 3* was made of pure bitumen.

The similarities and differences between the Dead Sea asphalt (*Ref 1*) and the black coatings of mummies are documented

with more precision by the analysis of the nuclear transitions of $^{14}$N. Among the three vanadyl porphyrin complexes VO-P1, VO-P2 and VO-P3 detected in *Ref 1* (but also in the commercial Judea bitumen *Ref 2*), VO-P1 and VO-P2 are also present in *Hum3*, while only VO-P1 was detected in the animal mummy *An 2*. As the latter contains less than 20% of the VO-P content of *Ref 1*, it is possible that VO-P2 peak intensity may be too low to be detected in this case. Bitumen from mummies *An 2* and *Hum 3* contain also a fourth complex VO-P4, that is absent in reference bitumen samples *Ref 1* and *Ref 2*. Instead, *Ref 1* and

*Ref 2* are characterized by the presence of VO-P3 complex. As the preparation of the embalming coating by ancient Egyptian implies that bitumen was heated to the liquid state in order to be mixed with the other ingredients and spread on the mummy, we may hypothesize that VO-P4 originates from the thermal transformation of an unstable complex VO-P3 initially present in natural bitumen. Laboratory experiments will be necessary to test this hypothesis.

**4 Conclusion and perspective**

In summary, this work shows that vanadyl porphyrin (VO-P) complexes commonly found at trace level in natural bitumen and oil can be used as intrinsic paramagnetic probes for a non-destructive analysis of the black coatings covering ancient Egyptian mummies and funerary artefacts. In a previous study by cw EPR (Dutoit, et al., 2020), we had shown that even small quantities of bitumen (relative to other organic substances) in these black coatings could be easily detected by the joint presence of VO-

P and carbon radicals ($C^0$) characteristic of fossil organic matter. Unlike conventional micro-destructive molecular analysis techniques (GC-MS), which often minimizes the presence of bitumen in black coatings (Lucejko, et al., 2017), EPR is non-destructive as samples are analyzed directly without preliminary physical or chemical treatment, and even a small amount of bitumen in a coating can be unambiguously detected.

In the present work, additional information on the nanostructure of the black coatings and on the speciation of vanadyl

porphyrins were obtained by hyperfine spectroscopy (ENDOR, HYSCORE). The $^1$H-ENDOR spectra reveal that the amplitude of the matrix line (representing distant hydrogen atoms) regularly increases with decreasing amount of bitumen in the black coating. This regular variation has been modeled, and indicates that all black coatings have similar nanostructures, with nano-sized aggregates of bitumen embedded in a matrix of bioorganic substances (conifer resin, fat, wax etc…). These similarities in nanostructures may reflect a similarity in preparation recipes of black coatings in various funerary contexts (animal and

human mummies, coffin) dating from the late period to the Greco-Roman period of the history of ancient Egypt. However, this hypothesis needs to be tested by studying a larger body of archaeological samples and by performing laboratory



reconstructions of black coatings. This should ultimately make it possible to specify the manufacturing recipes used in ancient Egypt. The speciation of VO-P was studied by detecting $^{14}$N nuclear transitions of porphyrins by HYSCORE spectroscopy. At least four types of VO-P complexes were identified from analysis of the double-quantum (dq-dq) correlation peaks of $^{14}$N.

These sharp peaks are relatively easy to detect even if the bitumen is mixed with other natural substances. Two of these VO-P complexes (VO-P1 and VO-P2) are present in both the reference bitumen samples and the coatings of the two human and animal mummies that we were able to study. The structure of a third type of VO-P present in reference bitumen samples (VO-P3), appears to have been transformed into another type (VO-P4) during the preparation of the coating of the two mummies. This indicates that some VO-P complexes may be thermally or chemically unstable, and their identification could give

information on the thermal/chemical treatments employed in ancient Egypt for the preparation of black coatings. This work and Dutoit et al. (2020) show that combining various EPR techniques (cw EPR, ENDOR, HYSCORE) is a promising tool for a non-destructive exploration of the nanostructure and composition of black coatings of ancient Egyptian mummies and funerary artifacts. This methodology should also permit to better apprehend local economies, workshop practices and recipes, supply areas as well as trade routes of bituminous materials in the past.


**Author Contributions.** CED, LB and OA recorded EPR and ENDOR spectra; CED and HV performed pulse-EPR measurements; DG interpreted the results. ALD carried out additional analyses by GC-MS; the manuscript was written by DG with contributions CED and LB. All authors have given approval to the final version of the manuscript.

**Competing interests.** One of the co-authors (HV) is member of the editorial board of Magnetic Resonance.

**Acknowledgment.** The authors are very grateful to the *Musée Dobrée* in Nantes, the *Musée d'Art et d'Histoire* in Narbonne, the *Musée des Confluences* in Lyon and the *Chateau-Musée de Boulogne-sur-Mer* and more especially to Julie Pellegrin, Camille Broucke, Flore Collette, Elikya Kandot and Gaëlle Etesse who are in charge of collections. In addition, we would like to express our gratitude to Nathalie Balcar, conservation scientist at the C2RMF for the mummy samples.

**Financial support.** CED, DG and LB received funding from *Agence Nationale de la Recherche* (ANR) under contract N° ANR-17-CE29-0002-01. Financial support from the IR INFRANALYTICS FR2054 for conducting the pulse EPR analyses is gratefully acknowledged.

**Supplementary information.** Samples; EPR spectra; ENDOR spectra; Derivation of Equation 1; HYSCORE spectra; Estimation of 2$^{nd}$ order contributions to the $^{14}$N parameters from dq-dq and sq-dq correlation peaks.




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
