# Peer review of "Insight into the structure of black coatings of ancient Egyptian mummies by advanced Electron magnetic resonance of vanadyl complexes."

_Magnetic Resonance, 2022_

## Author Comment (AC1)

**Point-by-point reply to referee # 1 for the manuscript:**

*Insight into the structure of black coatings of ancient Egyptian mummies by advanced Electron magnetic resonance of vanadyl complexes,* by CE. Dutoit, L. Binet, H. Fujii, A. Lattuati-Derieux and D. Gourier.

We sincerely thank the reviewer for his detailed reading of the manuscript, which allows us to correct an error and to address some points that need to be clarified.

**Comment #1:** *Line 134: It is very rare for organic radicals to have g values less than 2.002. Here, the authors give 1.9994. What is the uncertainty in this number ? How was the magnetic field calibrated ? Is the line asymmetric ?*

**Response:** Thank you for this comment, which is absolutely right. We have accurately measured the g-factors on these samples as well as very recently on 14 other samples from a new corpus of mummies. The vast majority of values are around $2.0037 \pm 2\times10^{-4}$, with a few samples deviating slightly from these values, $2.0032 \pm 2\times10^{-4}$ for the lowest, and $2.0042 \pm 2\times10^{-4}$ for the highest. The magnetic field was calibrated with a sample of DPPH ($g = 2.0037 \pm 2\times10^{-4}$). The organic radical line is mostly symmetrical both at X-band as well as at Q-band, indicating that the very slight asymmetry is not due to the anisotropy the g-factor (which is extremely small), but more probably to the influence of the neighbouring $m_I = +1/2$ and $-1/2$ hf lines of VO-P, which are very close to the $C^0$ line and may slightly distort the latter.
The g-value of radicals is now corrected in the manuscript (line 140, in red).

**Comment #2:** *It should be discussed why for the VO-nP species, not all hyperfine lines are visible. Presumably this is due to a distribution of hyperfine couplings resulting from ligation geometries that are less rigid than in porphyrins ?*

**Response:** In Figure 2, only the VO-nP lines that are most visible to the naked eye have been marked. Others can be seen by zooming in. There are two reasons why not all of them are visible: 1) several perpendicular transitions of VO-nP are hidden by VO-P lines; 2) the parallel lines are effectively broadened by a slight distribution of hyperfine couplings and g-factors (the broadening is more important at Q-band). However, simulations clearly have shown that VO-nP are vanadyls complexed by oxygenated ligands (Dutoit et al. 2020). The broadening of the parallel lines of VO-nP may indeed be due to a disorder effect linked to a much lower rigidity of the oxygenated ligands than that of the porphyrin rings.
A sentence has been added (lines 148-149, in red) concerning the apparently missing lines of the VO-nP

**Comment #3:** *In the experimental section, include the RF power used in the CW-ENDOR experiments.*

**Response:** The rf power in the ENDOR coil is 10 W. This value has been reported in line 106 of the manuscript.

**Comment #4:** *Figure 4c uses on the horizontal axis the weight-percentage VO-P content of the samples. Describe how this was determined. Also, add horizontal error bar to Figure 4c.*

**Response:** Each sample introduced into the EPR tube was carefully weighed, and the intensities were expressed per unit mass, $I_s/M_s$, where $I_s$ is the measured amplitude of a perpendicular hf line for the VO-P spectrum and $M_s$ is the mass of the historical sample. Taking the Dead Sea asphalt, consisting of pure bitumen, as a reference, for which $I_{ref}/M_{ref}$ is measured, the normalized intensities of the VO-P spectra (reported in wt%) of historical samples are $\dfrac{I_s\, M_{ref}}{I_{ref}\, M_s} \times 100$ (horizontal axis of Fig. 4c).

These details have been specified (in red) in the experimental part of the manuscript (lines 94-98).

The horizontal error bars are very small and correspond to the size of the data point. This is because the sample mass and amplitude of the hf line are measured with good accuracy. This information has been added in the caption of Fig.4c (in red).

**Comment #5:** *The X/Y ratio model assumes a Gaussian lineshape (see SI page 7, line 83). What is the assumed linewidth, and how is it justified. Also, why is the assumption of a Gaussian lineshape valid ?*

**Response:** In such amorphous solids with complex compositions and heterogeneous micro/nano-structures, and which are likely to have undergone weathering phenomena over time, a distribution of hyperfine couplings is naturally expected. Such distributions around a central value are generally Gaussian. A Gaussian line shape has therefore been postulated in the model developed in the SI. However, it is important to note that the conclusions of the model are not affected if another line shape is assumed. The only important thing is that the line shape is the same for all samples in the corpus, which is a reasonable assumption. A sentence has been added in the SI.

**Comment #6:** *Do the spectral features for A-parallel and A-perpendicular occur at the same frequency offset in all samples ?*

**Response:** Yes, the ENDOR features occur at the same frequency offsets in all samples of the corpus (this can be qualitatively seen in Fig. S5). Except for sample Hum 1, for which these ENDOR features are not detected because of the very low bitumen content, this indicates that the hyperfine coupling to the bridging hydrogens of the porphyrin rings is independent of the bioorganic content of the mixture, and thus that VO-P complexes are little affected by this mixture.

**Comment #8:** *Line 231: The statement that the A-parallel 1H ENDOR peak broadens and weakens as more bioorganic matter is mixed with bitumen needs some supporting evidence. Were reference spectra recorded for this ? It is obvious that disorder in the hyperfine coupling will broaden and weaken the peak, but from where is it known that the mixing of bitumen with other substances will lead to this ?*

**Response:** The weakening of the intensity (due to the decrease of the bitumen content of the mixture) and the broadening (due to the disorder) of the parallel components of the ENDOR spectra are just an experimental fact. This cannot be verified simply by making reference samples (for example by mixing bitumen and various natural substances), as this presupposes that we know the details of the recipes of the Egyptian priests (which was kept secret), which is precisely one of the objective of our work.

**Comment #9:** *Line 238: It is implied that protons more than 5-6 nm from the electron have zero electron-proton dipolar interactions. This is not correct. The interactions are small, but not zero.*

**Response:** Your remark is absolutely correct, but what we mean is that the dipolar interaction has no effect on the matrix ENDOR line shape and intensity beyond a certain distance r, even if this interaction is not zero. Kevan et al. (1976) have modelled the matrix lines in disordered systems, whose line shapes obey the expression:

$$f(v) = N \int\limits_{0}^{\pi} \int\limits_{a}^{\infty} \frac{\cos^2\theta \sin^2\theta}{r^4} \left[ \frac{1}{\alpha^2 + \left[v - (q/r^3)\right]^2} + \frac{1}{\alpha^2 + \left[v + (q/r^3)\right]^2} \right] d\theta dr$$

With *a* the average size of the unpaired electron wave function. They showed that $f(v)$ is almost independent of the electron-proton distance for *r* larger than a few nanometers.

Consequently, we have replaced "(limit for non-zero dipolar interaction)" by "(limit for an effect on the matrix line)" (line 247, in red).

**Comment #10:** *Line 265: It is stated that VO-nP complexes are localized to the interface between bitumen aggregates and natural substances. Could it also be that VO-P from bitumen has been solubilized as VO-nP during processing and has migrated into the non-bitumen phase ? How can this be excluded ?*

**Response:** Very good question, which we have been asking ourselves throughout this work. What is certain is that the vast majority of VO-Ps are intimately bonded to the asphaltene macromolecules of bitumen. This fraction is insoluble in organic solvents (but soluble in toluene). Thus, when all the organic components of the black matter of the historical samples are separated by treatments in n-alcanes and other solvents, only the insoluble asphaltene component precipitates, dragging with it the VO-Ps. In the historical samples, it is therefore highly doubtful that the VO-Ps can solubilize in the bioorganic component of the mixture. In a work in progress, we have recently measured electronic $T_1$ and $T_2$ for VO-Ps and $C^0$ radicals in these samples. Relaxation times $T_1$ and $T_2$ of VO-Ps do not vary across the series of historical samples and are the same as in pure natural bitumen ($T_1 \approx 1.5 \pm 0.5$ µs, $T_2 \approx 0.7 \pm 0.1$ µs), showing that the VO-Ps always "see" an asphaltene environment, whatever the bioorganic content of the samples. If the VO-Ps were more or less solubilized in the bioorganic component, variations in $T_1$ and $T_2$ would be expected across the series and would be different from values measured in pure bitumen. For these chemical and physical reasons, we assume that the trans-metalation reaction (formation of VO-nPs) takes place at the interface between asphaltene (which provides vanadyl ions) and the natural substances (which provide oxygenated ligands). However, as the black matter was viscous during the preparation of the mummies, it is likely that the VO-nPs, once formed by complexation with oxygenated functions of the bioorganic molecules, can subsequently diffuse into the bioorganic component before complete solidification of the coating. As the study of VO-nP is not the subject of this manuscript, we have chosen not to weigh down the text with these details.

**Comment #11:** *In Eq.(3), it is not clear what A∧(2) in the second term represents. Does the 2 indicate squaring or something else ?*

**Response:** This is probably a printing error on your side, because this symbol does not appear in the version available in Copernicus website. Equation 3 reads

$$v_{dq}^{\pm} = A \pm 2v_N + \frac{A^{(2)}}{(A/2) \pm v_N}$$

where $A^{(2)}$ is the second order contribution to the hyperfine interaction, given in Eq. S8.

**Comment #12:** *Regarding the orientation of the 14N quadrupolar tensor, what the DFT calculations predict ?*

**Response:** It is rather difficult to answer this question because DFT calculations give different predictions. Gracheva et al (2016) used the ORCA code to predict the $^{14}N$ ENDOR spectra for vanadyl porphyrins in crude oil. They obtained $Q_{zz}$ in the porphyrin plane, however their calculated ENDOR spectra differ somewhat from the experimental spectrum, especially for the magnetic field setting in the porphyrin plane. Gourier et al. (2010) used the Gaussian code to predict the $^{14}N$ HYSCORE spectrum of vanadyl tetraphenylporphyrin (VOTPP). They obtained $Q_{zz}$ nearly parallel to the VO bond, with simulated spectra in fairly good agreement with experimental spectra for both parallel and perpendicular field settings. In the present work, the experimental spectra can be fitted correctly only if $Q_{zz}$ is considered parallel to the VO bond, otherwise there is a factor of 2 between the value of $Q_{zz}$ obtained by fitting the spectra and the values of Q measured from sq-dq correlation peaks for the perpendicular field setting.

**Comment #13:** *By assessing different 14N peaks in the HYSCORE spectra to different VO-P species, it is implicitly assumed that all four nitrogens in a porphyrin complex has the same coupling parameters. Is this assumption valid ?*

**Response:** This is an important question, which may not be fully answered at present. On the one hand it is known that there are many varieties of vanadyl geoporphyrins in oil and bitumen, so they can be expected to differ in their $^{14}N$ hf coupling. On the other hand, VO-Ps with non-axial symmetry may exhibit different hf couplings for the fours nitrogen ligands. So either we are in the presence of only one (or two) type(s) of VO-P with different $^{14}N$ hf couplings, or we are in the presence of 4 different types of VO-Ps which are distinguished by the mean values of their $^{14}N$ hf coupling. The facts that the VO-P1 coupling is found in all 4 samples, while VO-P2 is found in only 3 samples, and that the 4 samples have either VO-P3 or VO-P4, but not both at the same time, favor the hypothesis of different types of VO-Ps rather than different types of nitrogen in a single VO-P. This is the choice we made.

**Other notes**

- *Line 69: In is unclear what a "hindered" hyperfine interaction is. Maybe "unresolved" hyperfine interaction is the intended meaning.* Right. This is now corrected in the manuscript.

- *Line 75: "Than in" → "as in"* Thank you. This is now corrected in the manuscript.

- *Figure S6: "chlorophyle" → "chlorophyll"* Yes, thank you. This is now corrected in the manuscript.

- *In general, // should be replaced by || to indicate "parallel*". It depends. We have always used this notation and it has always been accepted by reviewers and editors.

- *Line 104: Pulse EPR was probably done on an E580, not a E500 spectrometer.* Yes, thank you. This is now corrected in the manuscript.

- *Figure 3d: the three colors for the three wave packets indicating the three microwave pulses should be the same; different colors visually implies different frequencies.* The three colors indicate that these wave packets have a different role in the mechanism, but we can see that they have the same wavelength. In the case of ENDOR (Fig.3c), the rf line clearly has a wavelength larger than the microwave.

- *Line185: In cw ENDOR, the RF field doesn't have to be saturating.* We have removed "saturating" from the text. We used this word because the rf field changes the population of nuclear spin states.

- *Line 388: What is meant by "hindered" ? "hidden" ?* Yes, thank you for the remark.

---

## Author Comment (AC2)

**Point-by-point reply to referee # 2 for the manuscript:**

*Insight into the structure of black coatings of ancient Egyptian mummies by advanced Electron magnetic resonance of vanadyl complexes,* by CE. Dutoit, L. Binet, H. Fujii, A. Lattuati-Derieux and D. Gourier.

*This work presents an unusual application of EPR spectroscopy and related hyperfine techniques to the study of ancient egyptian mummies. The Authors apply ENDOR and HYSCORE spectroscopies to characterize the local environment of vanadyl porphirin complexes contained in the bitumen present in the embalming mixture. Based on different 14N hyperfine couplings, different types of vanadyl porphyrin complexes are identified, which appear to be related to the origin of the mummy. The work is well written and the experiments carefully performed and I think they represent a new and original application of hyperfine techniques. I have a few comments that should be addressed before publication.*

Thank you very much for this very positive appraisal of the work

**Comment #1:** *The different VO species are assigned based on small differences in the dq-dq correlation peaks shown in Figure 6. However, the spectra were recorded using a single tau value (tau=200 ns), which raises some doubts on the effect of blind spots. At least another tau value, possibly with a shorter length (96-100 ns) should be used to firmly assign the dq-dq cross peaks to different VO species and not, for example to the result of a distribution of hf values (strain). A simulation analysis showing the effect of different tau values on the dq-dq cross-peaks assuming the presence of the four VO different species or the effect of strain may be used as a possible alternative to substantiate the Authors assignment.*

**Response:** As the vanadyl porphyrin content of these natural and historical samples is low ($10^{17}$ to $10^{18}$ $V^{4+}$ per gram of black matter), the number of spins in the EPR tube was of the order of $10^{15}$ to $10^{16}$, which gives a low signal-to-noise ratio even in cw-EPR (see lines 18-20 of the revised manuscript, in red). Moreover, as a narrow region of the EPR spectrum is excited, the number of spins actually observed is extremely small. For these reasons it took us at least 20 hours to record each HYSCORE spectrum with an acceptable signal-to-noise ratio. It was therefore difficult to record spectra at different tau values. The value of 200ns was chosen from the 3P-ESEEM versus tau and 4P-ESEEM versus tau experiments on Dead Sea asphalt (Ref 1). The value of 200 ns was optimal for observing all the transitions. We repeated the same experiments for the mummy of unknown origin (Hum3) and this value was also canonical. For this sample we pushed a variation of tau up to 600ns and a value of 300ns was also good.

We have simulated the effect of tau on dq-dq spots as suggested by the reviewer (thanks for the suggestion). We simulated the HYSCORE spectra at tau = 100, 150, 200, 250, 300, 350 ns taking into account both VO-P1 and VO-P2 species. The resulting spectra in the dq-dq region are shown below and are reported also in the Supplementary Information Fig.S9 of the revised manuscript. It can be seen that the dq-dq of the two species remain well separated and well delimited at all values of tau.

[Figure]

*Effect of τ values on simulated dq-dq correlation peaks for VO-P1 and VO-P2 species, showing the lack of blind spot effects.*

**Comment #2:** *It would be interesting to compare the data from these unusual samples with those of related vanadyl phthalocyanine and porphyrin molecular complexes (see for example H. Moons, Z. Phys. Chem. 2017, 231, 887; K. Fukui, J. Phys. Chem. 1993, 97, 11858 and reference therein). In these systems the full 14N hyperfine and nuclear quadrupole tensors are derived. These values may be used as starting point for the simulation of the HYSCORE experiments.*

**Response:** Thank you for the suggestion. The paper by Moons et al. (2017) has been added in the references. The values measured "manually" ($a_{iso}$ =-7.3 MHz and $Q_{zz}$ = 0.94 MHz) and simulated ($a_{iso}$ = -7.3 MHz and $Q_{zz}$ = 1.0 MHz) for VO-P1 species in our samples are very close to the values $a_{iso}$ = -7.3 MHz and - 7.2 MHz for vanadyl tetraphenyl porphyrin (VOTPP) and vanadyl octaethyl porphyrin (VOEP) in toluene, respectively, with the corresponding values Q of 0.95 MHz and 1.00 MHz, respectively, for the quadrupolar interaction (measured by ESEEM by Fukui et al. 1993). The smaller $a_{iso}$ values measured for VO-P2 (-6.5 MHz) and for VO-P4 (-6.8 MHz) do not seem to have equivalent in the literature, however they are relatively close to those measured by Moons et al. (2017) in vanadyl perfluorophtalocyanine (-6.9 MHz). As phthalocyanines are synthetic molecules, we do not know whether they exist, even in very small quantities, in natural oils and bitumens (see lines 383-387 in the revised manuscript, in red).

**Comment #3:** *It would be useful to show at least as supplementary material, the simulation of the HYSCORE experiments taken at the two magnetic field settings superimposed to the experimental data. This allows to better judge on the quality of the simulation.*

**Response:** The superposition of an experimental spectrum and the simulated spectra at tau = 200 ns of VO-P1 and VO-P2 species is shown below and in Figure S10 for the Field setting $m_I$ = +3/2⊥ at 355.6 mT. Most of the features are reproduced. Concerning the field setting $m_I$ = -1/2 at 341.6 mT, this EPR transition suffers from a small, but not zero angular variation. However the excitation bandwidth of the HYSCORE spectrum being smaller than the EPR line width, it is not known which orientation range is selected by the first pulse. For this reason the comparison of the experimental and simulated spectra seems less relevant.

[Figure]

**Comment #4:** *The negative g-shift (g=1.9994) of the signal assigned to carbon organic radicals of asphaltene should be commented. Do the Authors have a structure in mind for these radical species?*

**Response:** Thank you for this comment, because the value reported in the manuscript is wrong. We have measured the g-factors on these samples as well as very recently on 14 other samples from a new corpus of mummies. The vast majority of values are around 2.0037, with a few samples deviating slightly from these values, 2.0032 for the lowest, and 2.0042 for the highest. The value 2.0037 (± 2x10$^{-4}$) is reported in the revised version of the manuscript. Such g-values are linked to the presence of heteroatoms (O, S) in the organic matter (Montanari et al. Appl. Magn. Reson. 14, 81, 1998).

Despite a very large number of EPR analyses of radicals in fossil organic matter (kerogen, coal, asphaltenes etc), there are currently no robust structural models for these radicals. They are characterized above all by their exceptional stability since they are also found in the organic matter of carbonaceous chondrites, the oldest objects in the solar system (~3.5 Ga) (Binet et al. 2002). These could be highly delocalised π-systems based on a phenalenyl-type structures, which are non-kekule structures with an odd number of carbons (Zhang et al. Energy & Fuels, 34, 9094, 2020). However their structure is unknown. The difficulty in proposing a precise structure is probably due to the presence of a distribution of a large number of radicals differing in size, structure and H/C ratio. This issue is challenging for advanced EPR techniques.

**Minor points**

1. In Table 2 it would be useful to include the assignment to the different VO-P complexes

It was not possible to attribute the EPR parameters in Table 2 to individual VO-Ps, as these natural bituminous materials contain a large number of different vanadyl porphyrins. The resulting EPR spectra are simply the sum of the individual spectra.

2. The value of the magnetic field setting at which the HYSCORE and ENDOR spectra have been recorded should be given in the Figures caption

The values of the magnetic field settings used in ENDOR and HYSCORE experiments are now given in captions of Figs. 4 and 5.